# Molecular Characterization and Phylogenetic Analysis of the Pine Tortoise Scale Insect *Toumeyella parvicornis* (Cockerell) (Hemiptera: Coccidae)

**Nicolò Di Sora** [1,†], **Silvia Turco** [1,*,†], **Federico Brugneti** [1], **Luca Rossini** [1,2,*], **Angelo Mazzaglia** [1], **Mario Contarini** [1] and **Stefano Speranza** [1,3]

1   Dipartimento di Scienze Agrarie e Forestali, Università degli Studi della Tuscia, Via San Camillo de Lellis snc, 01100 Viterbo, Italy; nico.disora@unitus.it (N.D.S.); federico.brugneti@unitus.it (F.B.); angmazza@unitus.it (A.M.); contarini@unitus.it (M.C.); speranza@unitus.it (S.S.)
2   Service d'Automatique et d'Analyse des Systèmes, Université Libre de Bruxelles, Av. F.D. Roosvelt 50, CP 165/55, 1050 Brussels, Belgium
3   Centro de Estudios Parasitológicos y de Vectores (CEPAVE, CONICET-UNLP), Boulevard 120 1900, La Plata B1900, Argentina
*   Correspondence: silvia.turco@unitus.it (S.T.); luca.rossini@unitus.it (L.R.)
†   These authors contributed equally to this work. Author order was determined in order of increasing seniority.

**Simple Summary:** Pest molecular characterization is an essential practice to support and complete the classical morphological taxonomy of insects, as well as to provide helpful information that may lead to finding alternative control strategies or detection methods. Molecular biology methods are of great importance, especially for the species whose identification requires highly qualified operators or the support of optical tools for microscopic organisms. This fact is even more amplified if the pest under study is an alien species that can quickly spread into a new area, as in the case of *Toumeyella parvicornis* (Cockerell) in Italy and Europe at large. This species demonstrates the need for molecular characterization, inspiring the present study.

**Abstract:** *Toumeyella parvicornis* (Cockerell) (Hemiptera, Coccidae) is becoming a potential main pest of stone pine plants (*Pinus pinea* L.), both in urban parks and pinewood forests in Europe. Its recent distribution is a source of concern in several regions of Italy and in France. The early detection of this pest plays a fundamental role to contain its geographical expansion, but its taxonomic identification is still based on laborious observations of the morphological traits. The identification is also complicated by the small size of the pest, which makes the observations possible only through a stereomicroscope. Molecular identification is beneficial for detection, but currently, there is only a single gene sequence available for this pest. This study fills this gap in knowledge by providing the sequences of five different genes (COI, 28S, elongation factor (EF-1$\alpha$), wingless (wg), and histone H3 (HexA)), together with a phylogenetic analysis carried out among species belonging to Coccidae, one of the most important families of scale insects. The results provide new valuable information about *T. parvicornis* and may represent useful data for its detection and management practices.

**Keywords:** pine tortoise scale; Coccidae; alien invasive species; molecular markers; phylogenetics; pest management; stone pine; urban forests

## 1. Introduction

The superfamily Coccoidea (Hempitera, Sternorrhyncha) includes organisms commonly known as scale insects [1] that feed on plant sap, causing serious damage and thus impacting agriculture and forest environments worldwide [2,3]. There are many examples of Coccoidea that have become serious threats in non-native areas, such as *Saissetia oleae* (Olivier) (Coccoidea, Coccidae), infesting plants of the genus *Olea* [4], or *Planococcus citri*

(Risso) (Coccoidea, Pseudococcidae) and *Parlatoria ziziphi* (Lucas) (Coccoidea, Diaspididae), infesting plants of the genus *Citrus* [5,6].

Among the scale insect group, the species belonging to the Coccidae family feed and develop on a plethora of host plants worldwide, from the northern to southern hemispheres [7]: *Ceroplastes rubens* Maskell infests, for instance, tea cultivations in some areas of Asia [8]; *Coccus hesperidum* L., is a highly polyphagous and globally distributed Coccidae pest [9]. Their invasive potential is enhanced by their small size, parthenogenetic reproduction, and by the absence of local natural enemies (i.e., predators, parasitoids, and diseases) in the new area of invasion [3,10,11]. Among the subfamily Myzolecaniinae, the genus *Toumeyella* surely deserves our attention [12,13], and in particular, the species *Toumeyella parvicornis* (Cockerell), commonly known as the tortoise scale insect. It is native to North America [7], where it feeds and develops on different species of the genus *Pinus* [14]. Recently, *T. parvicornis* has been introduced to Europe, where it is causing severe damages including tree diebacks in urban and natural *Pinus pinea* L. forests [15,16]. The main damage is caused by its feeding activity and the abundant production of honeydew, an optimal substrate for molds (Figure 1). The infestations in some areas of Central Italy, above all in the metropolitan area of Rome, are high to the point that the dieback of stone pines is changing the typical urban and coastal landscape [17]. The identification of *T. parvicornis* is still based uniquely on its morphological traits [16,18], preferably on the microscopic characteristics of the first nymphal stage (known as the crawler stage) or of the adult females [19,20]. As an alternative, it is possible to identify juvenile male stages through the design of the pupal covers [18].

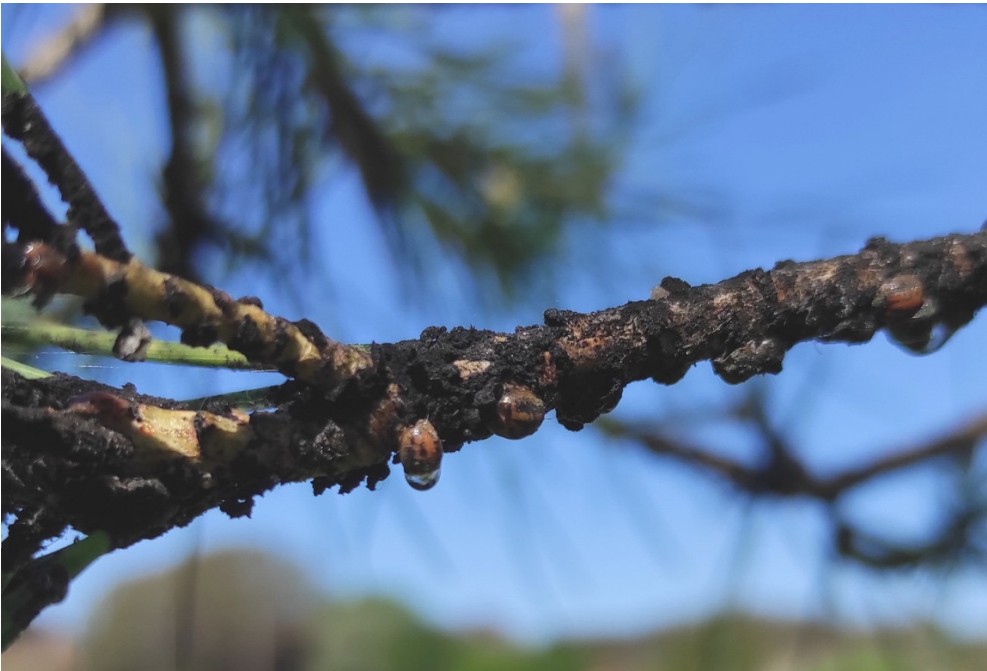

**Figure 1.** Mature adult females of *Toumeyella parvicornis* on stone pine twigs, with evidence of honeydew drops and black mold cover.

The genetic knowledge of this species is still scarce. Only two barcode sequences are currently available in the online GenBank databases, and both are related to the Cytochrome Oxidase Subunit I (COI) gene obtained by North American specimens [21]. Besides this little piece of information, at present, there is no genetic and molecular characterization of European specimens.

This work aims to fill this gap in the knowledge by providing the (partial) sequences of five different marked genes (COI, 28S, elongation factor 1α (EF-1α), wingless (wg), and histone H3 (HexA)) from DNA extracted by adult females of *T. parvicornis* collected in

Italy. These sequences were used to carry out a phylogenetic analysis with the Coccidae sequences available on the NCBI nucleotide database related to these five genes.

We believe that this information is of great importance since (i) it enriches the knowledge on the genetics of this species, making possible further comparisons between the strains adapted to the different areas worldwide; (ii) it lays the foundations for setting up faster detection methods based on molecular techniques; (iii) it lays the foundations for further studies on control methods; and (iv) it provides information on the introduction pathways, suitable for formulating more accurate containing strategies.

## 2. Materials and Methods

### 2.1. Collection of Toumeyella parvicornis Specimens and Morphological Identification

Specimens of *T. parvicornis* were collected in April 2023 from stone pines in an urban park located in Castel Fusano (Rome, Lazio, Italy, 41°44′29.8″ N, 12°19′44.1″ E, elevation: 0 m a.s.l.), within the facilities of the "Reparto Carabinieri Biodiversità di Roma". The sampling was carried out closely after the overwintering period of the adult females when there was a higher probability that the egg development had started. The stone pine plants were approximately 10 years old, with a diameter of 9 cm, a height of 3 m, circa, and spaced 6 m apart.

After the collection, specimens were sealed in plastic bags to avoid any dispersion and brought to the laboratory, where the first morphological identification was carried out under a stereomicroscope, following the illustrated keys of Hamon and Williams [7] and Miller and Williams [18], in the same way as Di Sora et al. [16]. The general characters of the specimens collected fit with the typical morphology of the Myzolecaniinae subfamily [1], in particular because of (i) the heavy sclerotization of the dorsum, (ii) anal plates with numerous setae, (iii) absence of the eyespot, (iv) progenital disc-pores with 5–6 loculi, (v) reduced antennae, and (vi) reduced legs. After this first rough selection, the specimens were morphologically characterized considering the fact that *T. parvicornis* adult females have dimensions of 3–3.5 mm in length and around 3 mm wide, a moderately elongated body shape, and a slightly sclerotized reddish-brown derm with dark spots (Figure 2a). The main characteristics for the identification of adult females are the typical presence of dorsal bilocular pore clusters evident on the dorsum region and the triangular anal plates (Figure 2b,c). Male pupal covers, instead, show a waxy surface that is rather fragile, an oblong shape with a slightly anterior elevation, the absence of relevant sculptures, and a single transverse posterior suture formed by the fusion of posterolateral and posterior transverse sutures (Figure 2d).

For the DNA extraction process, only the body content of the adult females was used (approximately 130–150 different individuals), mainly composed of eggs, in order to avoid the contaminated external part in the process. The dissection was carried out using two pins. The quantity of material extracted by the overall specimens collected was 450 mg. The dissected material was conserved in 1.5 mL Eppendorf tubes and stored in a fridge at 4 °C until DNA extraction.

### 2.2. DNA Extraction, PCR, and Sequencing

Total genomic DNA was extracted from the internal part of the body of adult females following a CTAB-based protocol. Briefly, 150 mg of material was incubated with 500 µL of lysis buffer (2% CTAB, 0.02 M EDTA, 0.1 M Tris-HCl pH 8, 1.2 M NaCl) at 65 °C for 30 min. One volume of chloroform:isoamyl alcohol (24:1) was added to the solution, vortexed, and centrifuged at 12,000 rpm for 10 min at room temperature. The aqueous phase was transferred to a new Eppendorf tube and further extracted with a second round of chloroform:isoamyl alcohol. The aqueous phase was then incubated with 10% volume of CTAB (10%) and 500 µL of additional chloroform:isoamyl alcohol. After centrifugation for 10 min at 12,000 rpm, the supernatant was transferred into a new Eppendorf tube and DNA was precipitated with one volume of cold isopropanol and 10% volume of 3M sodium acetate for 2 h at −20 °C. After 10 min of centrifugation at 12,000 rpm, the pellet

was washed with 70% ethanol and resuspended in 50 μL 65 °C preheated Tris-EDTA (TE). The DNA was quantified using the Invitrogen Qubit fluorometer (Thermo Fisher Scientific, Waltham, Massachusetts) and Optizen NanoQ spectrophotometer (Keen Innovative Solutions, Daejeon, Republic of Korea).

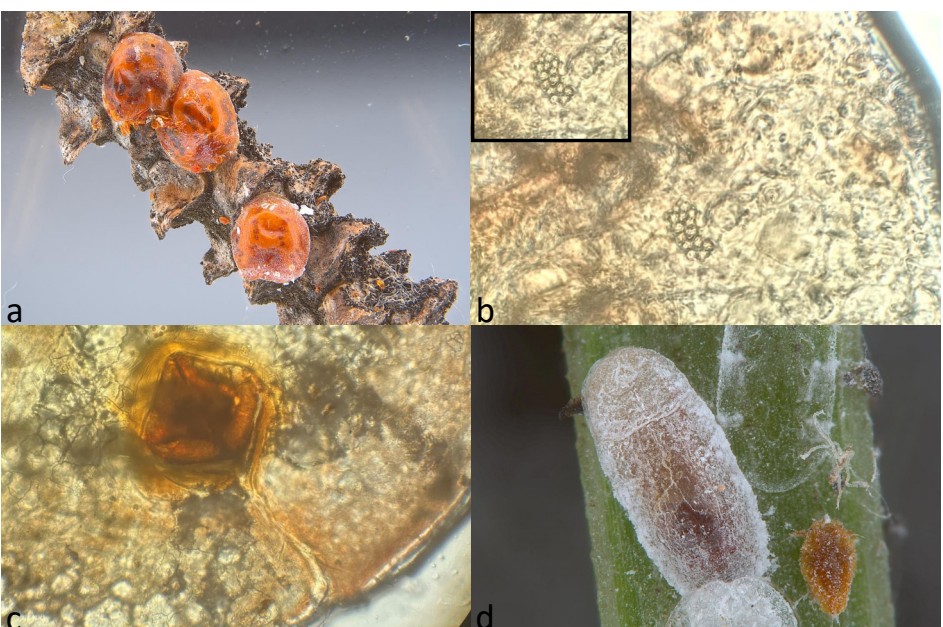

**Figure 2.** *Toumeyella parvicornis* identification characters: (**a**) adult females on stone pine twig, (**b**) slide-mounted female dorsal bilocular pore clusters, (**c**) slide-mounted female anal plates, (**d**) immature male cover.

Five different loci (COI, D2 region of the nuclear 28S gene, elongation factor (EF-1$\alpha$), wingless (wg), and histone H3 (HexA)) were selected for polymerase chain reaction (PCR) amplification using the primers listed in Table 1. The PCR reactions were prepared as follows: 2 μL of diluted DNA (~30 ng) was incubated with 0.4 μM of both forward and reverse primer and 1× of Ruby Taq Master mix (Jena Bioscience, Jena, Germany) in a final volume of 25 μL. PCRs were carried out using the GE-96G GeneExplorer Thermal Cycler (Bioer Technology, Hangzhou, China), with amplification programs adjusted according to the different primers. In particular, after an initial denaturation step at 95 °C for 15 min, 40 cycles were carried out at 94 °C for 30 s, 90 s of annealing at 48 °C (COI and EF-1$\alpha$), 55 °C (wg), 58 °C (28S), depending on the primers, elongation at 72 °C for 60 s and a final extension step at 72 °C for 10 min. The PCR products were visualized in a 1.5% agarose-TAE gel and then Sanger sequenced at Eurofins genomics (Eurofins Genomics GmbH, Konstanz, Germany). Their related electropherograms were inspected through FinchTV v1.4 (available at https://digitalworldbiology.com/FinchTV (accessed on 17 May 2023)) and their molecular identity was validated on NCBI nucleotide database with BLASTn and with the Barcode of Life Database System (BoldSystems V4) for the COI sequence [22]. Protein sequence alignment was carried out through Clustal Omega (available at https://www.ebi.ac.uk/Tools/msa/clustalo/ (accessed on 18 May 2023)).

*2.3. Phylogenetic Analysis*

The available sequences of the COI, 28S, wg, HexA, and EF-1$\alpha$ genes of the Coccidae family, belonging to Cardiococcinae, Ceroplastinae, Coccinae, Eriopeltinae, Eulecaniinae, Myzolecaniinae, and Filippiinae subfamilies, were downloaded from the NCBI nucleotide for the phylogenetic analysis (Table S1). The sequences were visually checked using UGENE [28] and aligned with MUSCLE v3.8.31 [29]. Phylogenetic relationships were reconstructed using Bayesian Inference (BI) methods with MRBAYES 3.2.7a [30]. The evolutionary model was set according to the standards suggested by the developers to a general

time-reversible model with gamma-distributed rates and a proportion of invariant sites ("GTR + I + Γ"). After some tuning attempts, the Markov Chain Monte Carlo simulation was run with 2,000,000 simulations, trees sampled every 100 generations, and diagnostics calculated every 1000 generations to achieve a standard deviation of split frequencies below 0.01. A maximum likelihood (ML) analysis using RAxML-HPC was carried out as well, using the GTRCATI algorithm as a substitution model and 1000 bootstraps [31]. The phylogenetic trees were visualized with FigTree v1.4.4 and the picture was edited with Inkscape v0.92 for visualization purposes (available at www.inkscape.org (accessed on 15 May 2023)).

**Table 1.** PCR primers used for molecular characterization of *Toumeyella parvicornis*.

| Primers | Sequence 5′-3′ | | Reference |
|---|---|---|---|
| | Forward | Reverse | |
| PCO-F1—LepR | CCTTCAACTAATCATAAAAATATYAG | TAAACTTCTGGAT GTCCAAAAAATCA | Amouroux et al. [23] |
| C-28SLong-F-C-28SLong-R | GAGAGTTMAASAGTACGTGAAAC | TCGGARGGA ACCAGCTACTA | Amouroux et al. [23] |
| M3-rcM44.9 | CACATYAACATTGTCGTSATYGG | CTTGATGAAAT CYCTGTGTCC | Cho et al. [24] |
| M44-t-rcM53.2 | CGAACGTGAACGTGGTATCAC * | GCAATGTGRGC [I]GTGTGGCA | Cho et al. [24] |
| scale_wg_F-LEPWG2 | CTGGTTCGTGCACGACGMGRACSTGYTGGATG | ACT[I]CGCARCACCAR TGGAATGTRCA | Hardy et al. [25] Brower & DeSalle [26] |
| H3 HexA-f-HexA-r | ATGGCTCGTACCAAGCAGACGGC | ATATCCTTGGGC ATGATGGTGAC | Zahniser et al. [27] |

* modified according to the *T. parvicornis* sequence, see text.

## 3. Results

Following the protocol described in the Materials and Methods section, 17.8 μg of genomic DNA was extracted from egg masses, with both 260/230 nm and 260/280 nm ratios falling between 1.7 and 2, and serially diluted for PCR amplifications. Amplification of the five different loci yielded sequences of the expected length: 585 bp for the COI, 825 bp for the 28S, 1101 bp for EF-1α, 362 bp for HexA, and 409 bp for wg. The sequences were deposited on the NCBI nucleotide database under the accession numbers: OQ996415, OQ991203, OR085314, OR004804, and OR004803, respectively.

### 3.1. Phylogenetic Relationships of the COI

A portion of the mitochondrial COI locus has been extensively used as a DNA barcode for insects of any kind thanks to a mutation rate fast enough to distinguish even between closely related species [32]. The partial COI sequence obtained here showed a percentage of identity of 91.40% and a query coverage of 97% to the only two corresponding COI sequences of *T. parvicornis* available on the NCBI blastn database and belonging to specimens isolated in Canada in 2010 (accession numbers KR041198 and HQ974643.1, respectively), followed by *Neolecanium cornuparvum* (Thro) with 88.24% of identity and 96% of query coverage. The translated amino acid sequence, instead, resulted in 95.45% and 94.44% identical to the two other *T. parvicornis*, with 92% of query coverage, followed by *Parthenolecanium corni* (Bouché) (85.22% identity, 94% coverage) and *N. cornuparvum* (93.43% and 92% coverage). Overall, the amino acid sequence is enriched in small and hydrophobic residues (in red), hydroxyl, sulfhydryl, or amine residues (in green), with only six basic (in magenta) and three acidic (in blue) residues (Figure 3).

For the analysis, we considered species belonging to the following: Cardiococcinae (3 sequences), Ceroplastinae (2 sequences), Coccinae (43 sequences), Eriopeltinae (3 sequences), Eulecaniinae (9 sequences), Myzolecaniinae (4 sequences), and Filippiinae (4 sequences), for a total of 68 species among the 7 subfamilies. Both Bayesian and ML inferred trees confirmed the close relationships obtained from BLASTn alignment, showing the three *T. parvicornis* and the *N. cornuparvum* (Myzolecaniinae subfamily) within the same

cluster and within the same branch with *Sphaerolecanium prunastri* (Boyer de Fonscolombe), *Lecanopsis formicarum* Newstead, and *Psilococcus ruber* Borchsenius, Eriopeltinae subfamily (Figures 4 and S1). In both trees, the rest of the closest branch, except for *Didesmococcus koreanus* Borchsenius (Eulecaniinae subfamily), *Ceroplastes cirripediformis* Comstock, and *Ceroplastes rubens* (Ceroplastinae subfamily), is composed of members of the Coccinae subfamily. Then, a separate further branch in the tree clusters with Eulecaniinae and Filippiinae, jointly with Cardioccinae and Coccinae members (Paralecaniini and Pulvinariini tribe species). A branch distant from the other ones, is composed by three Coccinae-Paralecaniini members (*Austrolecanium cryptocaryae* Lin & Cook, *A. sassafras* Gullan & Hodgson, and *Megalocryptes buteae* Takahashi), two Filippiinae members (*Ceronema banksiae* Maskell and *Austrolichtensia hakearum* (Fuller), and, just in the case of the Bayesian tree, one Eulecaniinae members (*Cryptes utzoni* Lin, Kondo & Cook).

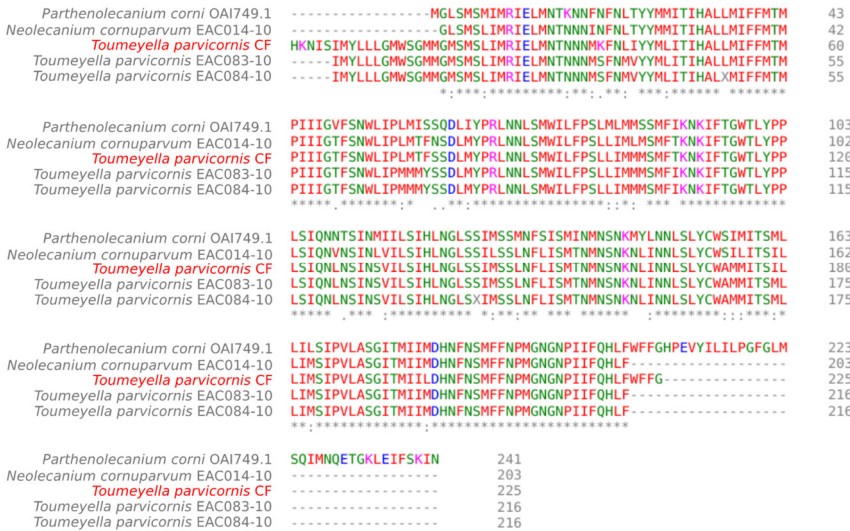

**Figure 3.** ClustalW multiple sequence alignment of the cytochrome oxidase subunit I (COI) protein sequence. Colors indicate the physicochemical properties: in red, small and hydrophobic residues; in blue, the acidic ones; in magenta, the basic residues; and in green, hydroxyl, sulfhydryl, and amine. The symbol "*" indicates perfectly match positions.

### 3.2. Phylogenetic Relationships of the 28S Ribosomal RNA Gene

The partial sequence of the 28S ribosomal gene was 88.4% identical to the 28S of *P. corni*, with a query coverage of 99%; 86.96% identical to *Coccus hesperidum*, with 99% of query coverage; and, among the others, 86.9% identical to *Coccus formicarii* (green), with 100% of coverage. For the analysis, we considered species belonging to the following: Cardiococcinae (2 sequences), Ceroplastinae (2 sequences), Coccinae (31 sequences), Eulecaniinae (7 sequences), Filippiinae (3 sequences), and Myzolecaniinae (1 sequence) subfamilies. To the best of our knowledge, no sequences are currently available from the closest relatives of *T. parvicornis*; thus, it clustered on its own in both BI and ML trees, but within the branch of Coccinae (mainly Paralecaniini tribe), Cardiococcinae, Eulecaninae, and Filippiinae species (Figures 5 and S2). Then, a separate, further branch in the tree was clustered with another Coccinae group (but mainly composed by Coccini tribe members) and Ceroplastinae.

### 3.3. Phylogenetic Relationships of the Wingless Gene

The *T. parvicornis* partial sequence of the wingless gene, necessary for the segment polarity and wing imaginal discs of insects, was, among others, 88.1% identical to the wg of *Coccus sulawesicus* Gavrilov, 87.23% to *C. formicarii*, and 87.78% to *C. discrepans* (green), with a query coverage of only 76%, 78%, and 78% of query coverage, respectively. For the analysis, we considered species belonging to the following: Coccinae (17 sequences) and Myzolecaniinae (1 sequence). Even in this case, no sequences are currently available from the closest relatives

of *T. parvicornis*; accordingly, it clustered on its own in both BI and ML trees but within the branch of Coccinae (all species belonging to the Coccini tribe) (Figures S3 and S4).

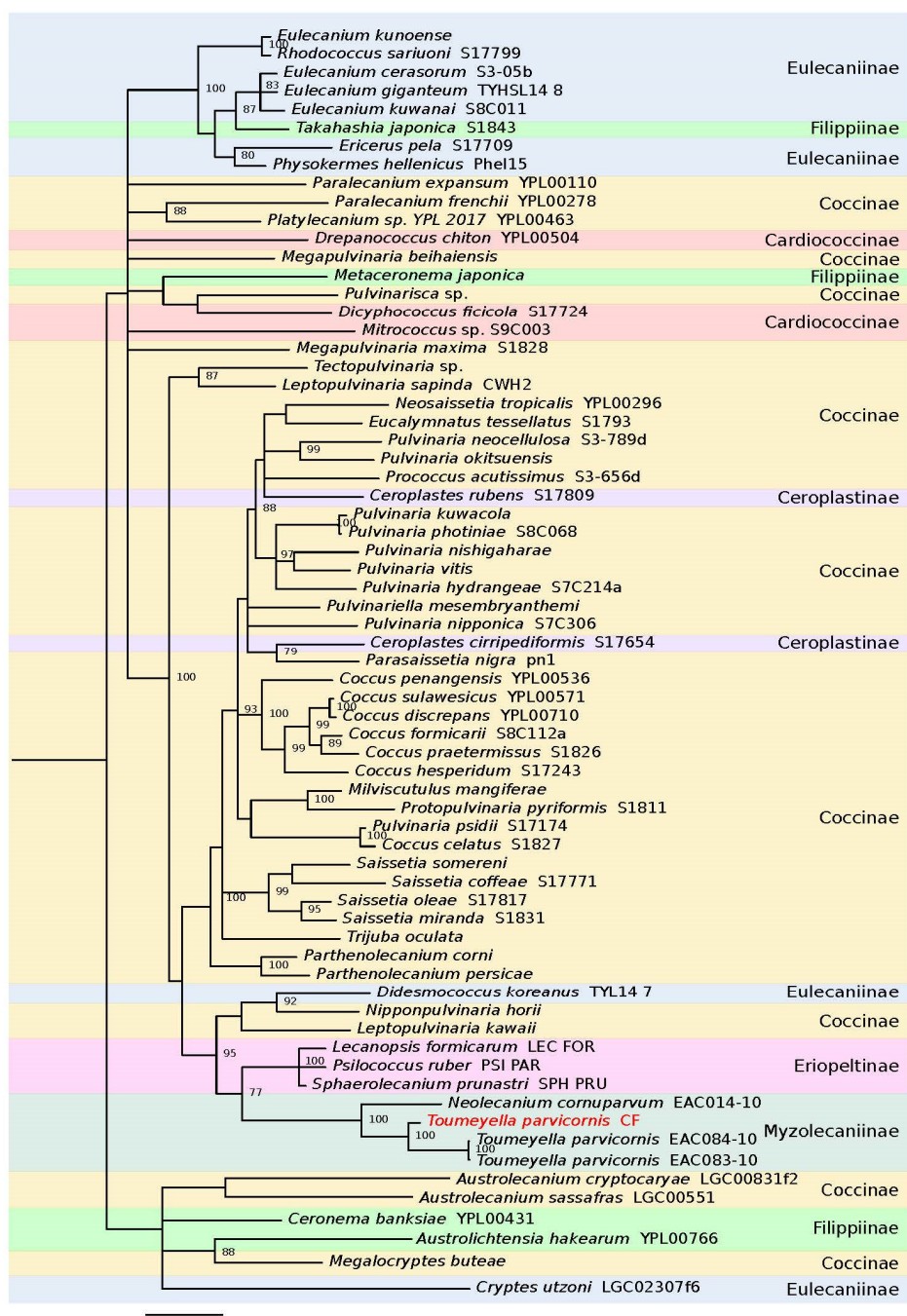

**Figure 4.** MrBayes phylogenetic tree of the cytochrome oxidase subunit I (COI) among the Coccidae family. The colors indicate the morphological subfamilies: Eulecaniinae, Filippiinae, Coccinae, Cardiococcinae, Ceroplastinae, Eriopeltinae, and Myzolecaniinae, to which *T. parvicornis* belongs. In red is the Italian isolate of *T. parvicornis*. The numbers in the branch labels indicate the percentage of Bayesian posterior probability.

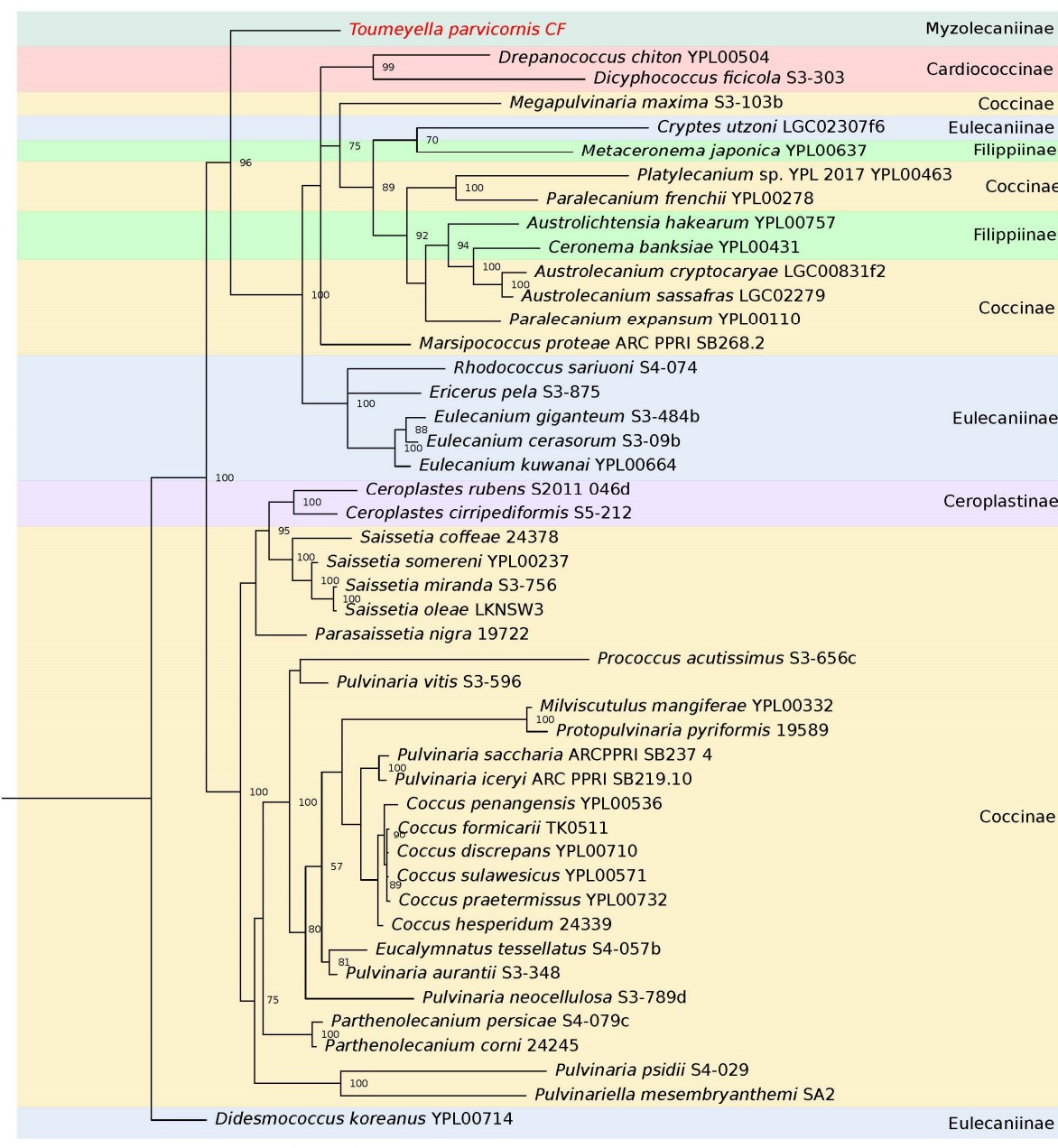

**Figure 5.** MrBayes phylogenetic tree of the rRNA 28S fragment D gene among the Coccidae family. Colors indicate the morphological subfamilies: Eulecaniinae, Filippiinae, Coccinae, Cardiococcinae, Ceroplastinae, Eriopeltinae, and Myzolecaniinae, to which *T. parvicornis* belongs. In red is the Italian isolate of *T. parvicornis*. The numbers in the branch labels indicate the percentage of Bayesian posterior probability.

### 3.4. Molecular Characterization of the Elongation Factor 1-Alpha

The primer pair used for the first amplification of the EF-1α yielded a product of 343 bp related to the 5′ end of the gene, which included an intron fragment [33]. However, this product did not align with the other EF-1α sequences of Coccidae available on the NBCI database because their sequences were obtained using downstream oligonucleotides. Indeed, when blasted towards the Coccidae database, the amplified 343 bp product aligned only to *Ceroplastes* sp. IMV-2016 EF-1α with 80.18% identity and a coverage of 95% and to *Eulecanium kunoense* (Kuwana) with 83.95% identity and a coverage of only 47%. Thus, the M44-1 primer from Cho et al. [24] was modified according to the 5′ end sequence of

*T. parvicornis* and used in combination with rcM53.2 to amplify a final product of 1101 bp. Part of this long sequence was then aligned to members of the Coccidae family (*Ericerus pela* (Chavannes), *Platylecanium* sp., *Saissetia oleae*), with whom shared an average percentage of identity of 77% and a coverage of 54% at the nucleotide level, in order to obtain the Bayesian and ML trees shown in Figures 6 and S5. *Toumeyella parvicornis* clustered in a branch close to Coccinae species (mainly belonging to Paralecaniini and Pulvinariini tribes), Cardiococcinae, Eulecaniinae, and Filippiinae.

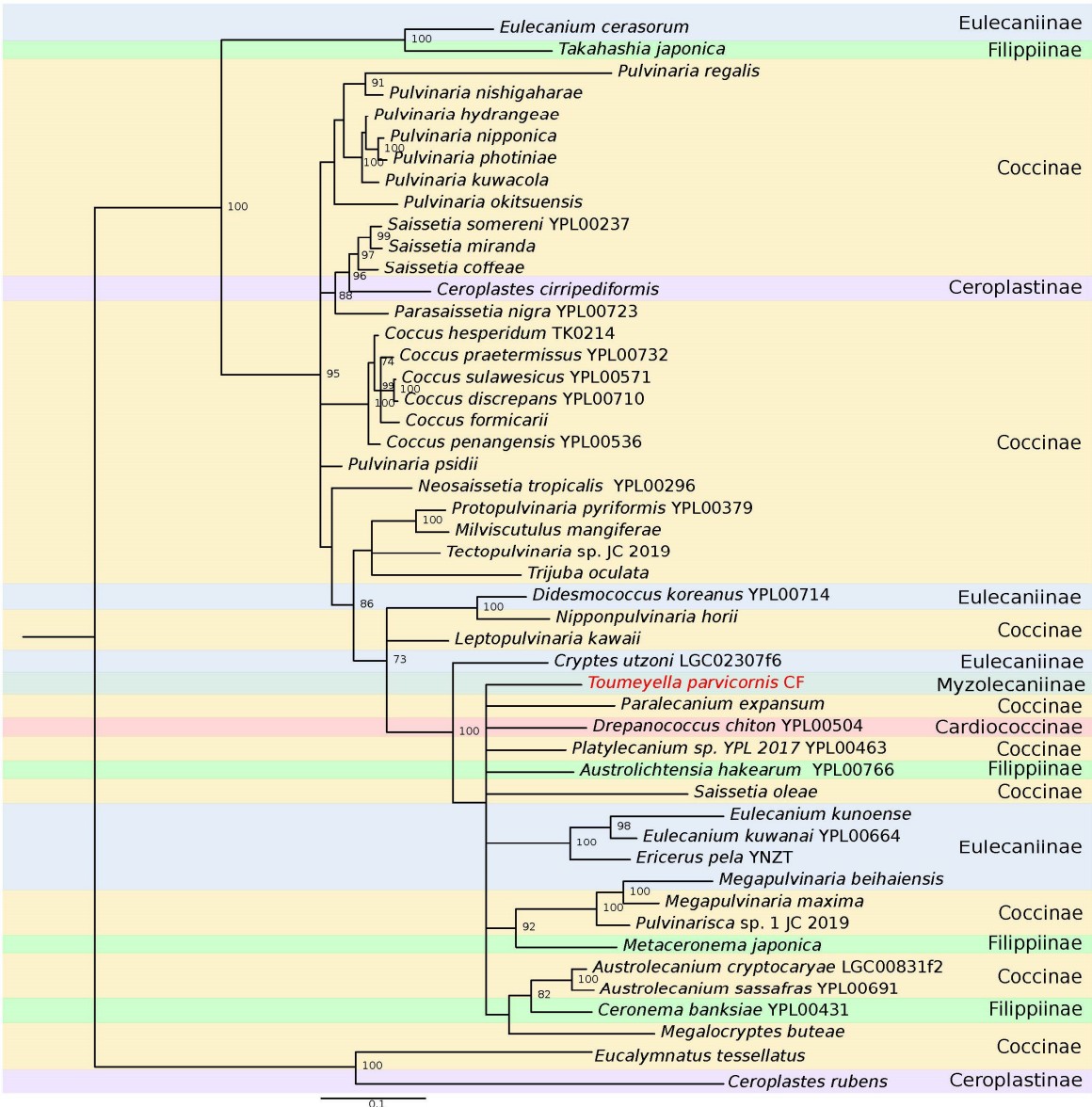

**Figure 6.** MrBayes phylogenetic tree of the elongation factor EF-1α gene among the Coccidae family. Colors indicate the morphological subfamilies: Eulecaniinae, Filippiinae, Coccinae, Cardiococcinae, Ceroplastinae, Eriopeltinae, and Myzolecaniinae, to which *T. parvicornis* belongs. In red is the Italian isolate of *T. parvicornis*. The numbers in the branch labels indicate the percentage of Bayesian posterior probability.

*3.5. Genetic Characterization of the Histone H3*

The phylogenetic tree of HexA among the Coccidae could not be obtained, given that only three sequences are currently available on the NCBI database among Coccidae, Pseudococcidae, and Monophlebidae families. Furthermore, no similarity was found between the HexA sequence of *T. parvicornis* and *P. citri* of the family Pseudococcidae (accession

number AM409515). Instead, the sequence is 85.93% identical, with 90% coverage to *Pulvinaria regalis* Canard (accession number GU066931, family Coccidae) and 82.37% identical, with 99% of coverage to a *Crypticerya* sp. isolate (EU087882, family Monophlebidae).

## 4. Discussion

The molecular phylogeny of insects based on a reference collection of DNA sequences obtained from individuals of reliably identified species was used for the first time more than 30 years ago [34]. Since then, this approach has progressively gained significance and reliability, and the number of "barcode" sequences has exponentially increased, so that in 2020, Chesters was able to infer the phylogeny of 69,000 species [35]. To date, molecular data are widely used to decrypt the complex taxonomy of insects. An example somehow related to the taxa involved in this study was published by Normark and colleagues [36], who were able to resolve the taxonomic classification of the armored scale insects belonging to Diaspididae by flanking a careful morphological identification to the molecular phylogeny obtained by concatenating COI and COII, EF-1$\alpha$, 28S, and 16S gene sequences.

The present study reports the first molecular and phylogenetic analysis of European specimens of *T. parvicornis* since its first detection in 2014 [19]. The results provide the partial sequence of five marker genes (COI, 28S, EF-1$\alpha$, wg, and HexA), four of which were sequenced for the first time. As a reference point for our study, we have considered COI because it is commonly deemed the insect barcoding gene. Additionally, a sequence of the same gene was already available for *T. parvicornis*, even if extracted from different specimens. In our results, the COI phylogenetic analysis showed a relationship between the Italian *T. parvicornis* and the two specimens from North America, together with *N. cornuparvum*. These results corroborate the hypothesis that *T. parvicornis* and *N. cornuparvum* are part of two controversial genera, *Neolecanium* being considered a synonym, by some authors, of the *Toumeyella* genus [1,37]. Some examples of strong similarities are already represented by the case of the species *N. leucaenae* (Cockerell), *Toumeyella cerifera* (Ferris), and *T. sonorensis* (Cockerell) that are now considered to be part of a common new genus named *Neotoumeyella* Kondo & Williams [38].

Furthermore, the COI tree revealed phylogenetic affinities between species that are partially in accordance with the classical morphologically based classification of Coccidae proposed by Miller and Hodgson [39] and the phylogenesis conducted by Choi and Lee [40]. In our analysis, the Myzolecaniinae taxon was more related to the Eriopeltinae and Coccinae groups, and, in line with Choi and Lee [40], we found that the Eulecaniinae subfamily is related to the Filippinae group as well, and Cardiococcinae and Ceroplastinae share similarity with the Coccinae group. As originally assumed by Miller and Hodgson [39] and further confirmed by Lin et al. [41] and Choi and Lee [40], the Coccinae group requires a taxonomic revision, especially regarding the Paralecaniini and Pulvinariini tribes, due to their clear distance from the rest of the Coccinae tribes. Indeed, our results further showed a separation of *Austrolecanium cryptocaryae*, *A. sassafras*, and *Megalocryptes buteae* of the Paralecaniini tribe from the Coccinae main group by clustering together with members of the Eulecaniinae, Filippiinae, and Cardiococcinae subfamilies.

In the 28S gene phylogenetic trees, two main groups are formed: one with Coccinae and Ceroplastinae subfamilies, in line with the phylogenetic association previously described [23,41], and a second branch with Eulecaniinae, Coccinae, Filippiinae, Cardiococcinae subfamilies, and the only Myzolecaniinae member, *T. parvicornis*. Interestingly, the phylogenetic tree built on the sequences of the EF-1$\alpha$ showed the same clustering.

Wg results confirmed the affinities evident in the COI tree, even if only with the Coccinae subfamily, and in particular with members belonging to the Coccini tribe, since wg gene sequences of other subfamilies and species were not available on NCBI database. The HexA gene comparison was only possible between four species, with only one of them belonging to the Coccidae family, and precisely *P. regalis* (Coccidae, Coccinae) showed the highest similarity.

Besides phylogeny, it is worth noting that the molecular approach can provide precious information also about the evolutionary history of entire taxonomic groups [42].

*Toumeyella parvicornis* represents a serious phytosanitary emergence, considering its high impact on its host plant, *P. pinea* [13]. The major risk is linked to a radical change in the coastal landscape in the Mediterranean basin, where stone pine is widely distributed [43]. Because of this potentially devastating environmental impact, several attempts have already been made to better understand and control this pest [16,44], but the more powerful control strategies are probably "written in their genetic code" [45]. Indeed, more than 95% of the sequences deposited on the BOLD system database were unique and highly distinctive in the COI sequence [22,32]. DNA barcoding using the COI sequence allowed the identification of new species, especially when morphological characterization is challenging or could lead to species misidentifications [46]. DNA barcoding deserves to be improved to enhance pest management in the coming years [47,48], since genetic tools have already been applied as biosurveillance strategies, especially for alien pest management [45]. As an example, recently COI barcoding has been applied to discriminate mealybugs (family Pseudococcidae) affecting coffee plants in Brazil in order to develop integrated pest management (IPM) and tempestive quarantine interceptions [46]. Interestingly, fast detection methods can be easily derived from molecular data used in phylogeny. As an example, a specific PCR-based assay was developed on the 28S ribosomal gene sequence of the species *Aspidiotus rigidus* Reyne (Hempitera: Diaspididae), the false coconut scale, allowing a quick and affordable detection method capable of discriminating it from the other species of the same genus retrievable in coconut plantations in the Philippines and providing a key tool to farmers for appropriate and timely management of this pest [49].

Moreover, the analysis of the sequences of marker genes can be used to describe the genetic structure of different populations of a pest species during an outbreak, as for such as COI and EF-1$\alpha$ for the same *A. rigidus* in the Philippines [50], providing valuable information about the pathways of their dispersal. Considering *T. parvicornis* potential expansion scenario [44], the newly deposited sequences may already be useful to accelerate its detection and control strategies. Also, further genetic studies on specimens of *T. parvicornis* from the different areas of presence worldwide should allow us to track its origin and the likely incoming routes. This information, when available, will be helpful to explain why this pest adapted and spread quickly in the Italian peninsula, how many points of entry the species had, and how its expansion can be prevented [51]. For example, the molecular approach could be used to explain the spread of this pest in Italy, where the first Italian outbreak was reported in the Naples area [19] and almost independently in the surrounding area of Rome, towards the seaside [16,17]. To date, in fact, there is not a diffusion pathway that connects the two areas, and only an investigation of the genetic relationship between the individuals of the two populations may explain what really happened. This methodology has already proven to be effective in other studies, as, for example, for *Halyomorpha halys* (Stål) (Hemiptera, Pentatomidae), where the entry pathways have been determined through the identification of different populations [52]. Moreover, the number of yearly generations that the insect completes in a Mediterranean climate should raise an additional question. Given the presence of the pest in North and Central America and the diversity of the climate in these areas, we may ask ourselves if *T. parvicornis* adapted quickly to Central Italy because the area of provenance had similar environmental conditions.

The answers to these open-ended questions would be very helpful in reinforcing the problematic control of incoming commercial materials and becoming a landmark case study in managing the spread of similar insect pests.

## 5. Conclusions

This molecular study provides five new gene sequences of the Italian population of *T. parvicornis*. The COI sequence matched the only other available COI sequences of *T. parvicornis* extracted from North American specimens, while the other four sequences are

completely new to science. Phylogenetic analysis confirmed the belonging of *T. parvicornis* to the Myzolaecaninae subfamily and its closeness to the Coccinae subfamily. The results presented in this work may simplify the identification practices of the pest in urban and natural stone pine areas and open the door to future genetic evaluations of the populations to better understand the pathways of invasion.

**Supplementary Materials:** The following supporting information can be downloaded at https://www.mdpi.com/article/10.3390/f14081585/s1, Figure S1: RAxML phylogenetic tree of the cytochrome oxidase subunit I (COI) among the Coccidae family. The colors indicate the morphological subfamilies: Eulecaniinae, Filippiinae, Coccinae, Cardiococcinae, Ceroplastinae, Eriopeltinae, and Myzolecaniinae, to which *T. parvicornis* belongs. In red is the Italian isolate of *T. parvicornis*. Bootstraps values are indicated in the branch labels; Figure S2. RAxML phylogenetic tree of the rRNA 28S fragment D gene among the Coccidae family. Colors indicate the morphological subfamilies: Eulecaniinae, Filippiinae, Coccinae, Cardiococcinae, Ceroplastinae, Eriopeltinae, and Myzolecaniinae, to which T. parvicornis belongs. In red is the Italian isolate of *T. parvicornis*. Bootstraps values are indicated in the branch labels; Figure S3: MrBayes phylogenetic tree of the wingless gene among the Coccidae family and *T. parvicornis*. The numbers in the branch labels indicate the percentage of Bayesian posterior probability; Figure S4: RAxML phylogenetic tree of the wingless gene among the Coccidae family and *T. parvicornis*. The numbers in the branch labels indicate the bootstrapt values; Figure S5: RAxML phylogenetic tree of the elongation factor EF-1$\alpha$ gene among the Coccidae family. Colors indicate the morphological subfamilies: Eulecaniinae, Filippiinae, Coccinae, Cardiococcinae, Ceroplastinae, Eriopeltinae, and Myzolecaniinae, to which *T. parvicornis* belongs. In red is the Italian isolate of *T. parvicornis*. The numbers in the branch labels indicate the bootstrap values; Table S1: List of the species and their related accession numbers used for the phylogenetic analysis.

**Author Contributions:** Conceptualization, N.D.S., S.T., F.B., L.R., A.M., M.C. and S.S.; methodology, N.D.S., S.T., F.B. and L.R.; software, S.T. and L.R.; formal analysis, N.D.S., S.T., F.B. and L.R.; investigation, N.D.S., S.T. and F.B.; resources, A.M., M.C. and S.S.; data curation, N.D.S., S.T. and L.R.; writing—original draft preparation, N.D.S. and S.T.; writing—review and editing, N.D.S., S.T., F.B., L.R., A.M., M.C. and S.S.; visualization, N.D.S., S.T. and L.R.; supervision, A.M., M.C. and S.S.; project administration, M.C. and S.S.; funding acquisition, S.S. All authors have read and agreed to the published version of the manuscript.

**Funding:** N.D.S. is funded by the Lazio Region (Agriculture Department) and the Università degli Studi della Tuscia (Italy). S.T. is funded by the Italian MIUR (Ministry of Education, University and Research) initiative 'Department of Excellence' (Law 232/2016). L.R. is funded by Italian MUR (Ministry of University and Research) in the framework of the European Social Funding REACT-EU—National Program for the Research and Innovation 2014–2020 and by the European Commission under the Grant n. 101102281, Project "PestFinder", call HORIZON-MSCA-2022-PF-01.

**Data Availability Statement:** The nucleotide sequences related to this work have been deposited and are available on the NCBI database under the accession numbers: OQ996415, OQ991203, OR085314, OR004804, and OR004803.

**Acknowledgments:** The authors are grateful to the two reviewers for their helpful comments and suggestions that have been greatly helpful for improvements to this manuscript. The authors are grateful to "Reparto Carabinieri Biodiversità di Roma", in particular to Danilo Bucini and Roberta Zini for their support, and to Simone Menegoni for the support during field activity. All the bioinformatics calculations and analyses were carried out at the DAFNE HPC scientific computing centre of the Università degli Studi della Tuscia.

**Conflicts of Interest:** The authors declare no conflict of interest.

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
