# Peer review of "Molecular Characterization and Phylogenetic Analysis of the Pine Tortoise Scale Insect Toumeyella parvicornis (Cockerell) (Hemiptera: Coccidae)"

_forests, doi:10.3390/f14081585_

Round 1
Reviewer 1 Report
The paper deals with the scale insect Toumeyella parvicornis (Cockerell) (Hemiptera: Coccidae). Although no morphological evidence is presented except only one photo of low quality, where hardly any distinguishing features are visible. I am sure the paragraph with short morphological characterization is needed with some graphic of photographic images. Otherwise there is no evidence that exactly this species was molecularly studied. It is also essential to give short description of peculiar characters of Myzolaecaninae subfamily and discuss whether morphological and molecular results are congruent or not.
Introduction (paragraph 3)
written now: "… local natural defenders (i.e. predators, parasitoids, and diseases)"
The term "natural defenders" is unknown to me and other specialists, must be natural enemies.
Author Response
Response Reviewer 1
Reviewer 1:
The paper deals with the scale insect Toumeyella parvicornis (Cockerell) (Hemiptera: Coccidae). Although no morphological evidence is presented except only one photo of low quality, where hardly any distinguishing features are visible. I am sure the paragraph with short morphological characterization is needed with some graphic of photographic images. Otherwise there is no evidence that exactly this species was molecularly studied. It is also essential to give short description of peculiar characters of Myzolaecaninae subfamily and discuss whether morphological and molecular results are congruent or not.
Response:
Dear Reviewer 1, thank you for the time dedicated to revise our manuscript, as well as for the helpful comments and suggestions provided. We sincerely appreciate your positive comments, and we carefully addressed all the issues raised. For the reader’s convenience, we added a paragraph reporting the most important traits of the Myzolaecaninae subfamily, completed by references, as well as some pictures that help the morphological characterization of the specimens.
We hope that this change fits with your expectations, and that now the overall quality of the manuscript, as well as its readability, has been sufficiently improved. We renew our availability for any further change or request, if needed.
Thank you again.
Reviewer 1:
Comments on the Quality of English Language
Introduction (paragraph 3)
written now: "… local natural defenders (i.e. predators, parasitoids, and diseases)"
The term "natural defenders" is unknown to me and other specialists, must be natural enemies.
Response:
Thank you for this comment. We changed this word accordingly to use a language that is as familiar as possible to the readers.
Reviewer 2 Report
In the present study, the authors have developed a molecular characterization method together with a phylogenetic analysis to early detect Toumeyella parvicornis (Cockerell), as the potential main pest on stone pine plants. Their findings will greatly contribute to provide new valuable information for T. parvicornis detection and it management practices. The methods, results and conclusions are scientifically sound. However, this manuscript has some major and minor concerns as currently written that should be addressed.
Major concerns:
1. The authors should compare and discuss the advantage and disadvantage of molecular detection for T. parvicornis using five marker genes including COI, 28S, EF-1α, wg and HexA to provide more scientific references for the molecular identification of other similar pests.
2. I strongly recommend the authors should abbreviate cytochrome oxidase subunit I as COI according to the common abbreviation rules. In addition, the full name and its corresponding abbreviation should be used correctly.
Minor concerns:
Please revise them in the attachment carefully.

This manuscript should require moderate editing of English language.
Author Response
Response Reviewer 2
Reviewer 2:
In the present study, the authors have developed a molecular characterization method together with a phylogenetic analysis to early detect Toumeyella parvicornis (Cockerell), as the potential main pest on stone pine plants. Their findings will greatly contribute to provide new valuable information for T. parvicornis detection and it management practices. The methods, results and conclusions are scientifically sound. However, this manuscript has some major and minor concerns as currently written that should be addressed.
Response:
Dear Reviewer 2, thank you for the time dedicated to revise our manuscript, as well as for the helpful comments and suggestions provided. We sincerely appreciate your positive comments, and we carefully addressed all the issues raised. In particular, we have followed all the suggestions on the attached PDF and fixed all the issues with COI abbreviation and scientific names (and abbreviations) of the species.
We hope that this change fits with your expectations, and that now the overall quality of the manuscript, as well as its readability, has been sufficiently improved. We renew our availability for any further change or request, if needed.
Thank you again.
Reviewer 2:
Major concerns:
- The authors should compare and discuss the advantage and disadvantage of molecular detection for T. parvicornis using five marker genes including COI, 28S, EF-1α, wg and HexA to provide more scientific references for the molecular identification of other similar pests.
Response:
The main aim of the paper was to provide scientific evidence of the effectiveness of molecular methods in assessing the correct taxonomic position of this insect species. We believe that this task was fully accomplished on one side, by confirming the correspondence between the two sequences of COI from T. parvicornis deposited in NCBI and those obtained here, and, on the other hand, by providing brand new sequences of housekeeping genes for the same species, which will be of great help for a correct taxonomic collocation of future specimens. About molecular detection, even if it is not a specific target of this study, it surely paves the way for the depiction of a specific molecular assay (standard PCR or qPCR based) to be used for fast identification of putative T. parvicornis specimens, provided that suitable sequences for the design of specific primers or probe can be found.
Aiming to better explain these concepts and fulfill your request, a sentence was added to the Discussion paragraph. Also, examples of the usefulness of these sequences in both phylogeny and detection assay for other insect pests are added.
Reviewer 2:
- I strongly recommend the authors should abbreviate cytochrome oxidase subunit I as COI according to the common abbreviation rules. In addition, the full name and its corresponding abbreviation should be used correctly.
Response:
Thank you very much for this suggestion. We paid attention to correct the abbreviation of COI, as well as to abbreviate correctly the names of the species depending on the sentences and of the part of the text. We hope that the changes fit with your expectations.
Reviewer 2:
Minor concerns: Please revise them in the attachment carefully.
Response:
Thank you for suggesting us stylistic corrections and helpful advice to improve the readability of the manuscript. We have faithfully reported all the corrections on the revised version of the manuscript. We hope that this change fits with your expectations.